# Case Study on the Fitting Method of Typical Objects

**Liu Zhang** [1,2], **Jiakun Zhang** [1,2], **Hongzhen Song** [1], **Wen Zhang** [1] **and Wenhua Wang** [1,2,*]

1   College of Instrumentation& Engineering Electrical, Jilin Universuty, Changchun 130000, China; zhangliu@jlu.edu.cn (L.Z.); jkzhang20@mails.jlu.edu.cn (J.Z.); songhz19@mails.jlu.edu.cn (H.S.); zhangwen19@mails.jlu.edu.cn (W.Z.)

2   National Eophysical Exploration Equipment Engineering Research Center, Jilin Universuty, Changchun 130000, China

*   Correspondence: wangwh900@jlu.edu.cn

**Abstract:** This study proposes different fitting methods for different types of targets in the 400–900 nm wavelength range, based on convex optimization algorithms, to achieve the effect of high-precision spectral reconstruction for small space-borne spectrometers. This article first expounds on the mathematical model in the imaging process of the small spectrometer and discretizes it into an *AX=B* matrix equation. Second, the design basis of the filter transmittance curve is explained. Furthermore, a convex optimization algorithm is used, based on 50 filters, and appropriate constraints are added to solve the target spectrum. First, in terms of spectrum fitting, six different ground object spectra are selected, and Gaussian fitting, polynomial fitting, and Fourier fitting are used to fit the original data and analyze the best fit of each target spectrum. Then the transmittance curve of the filter is equally divided, and the corresponding *AX=B* discrete equation set is obtained for the specific object target, and a random error of 1% is applied to the equation set to obtain the discrete spectral value. The fitting is performed for each case to determine the best fitting method with errors. Subsequently, the transmittance curve of the filter with the detector characteristics is equally divided, and the corresponding *AX=B* discrete equation set is obtained for the specific object target. A random error of 1% is applied to the equation set to obtain the error. After the discrete spectral values are obtained, the fitting is performed again, and the best fitting method is determined. In order to evaluate the fitting accuracy of the original spectral data and the reconstruction accuracy of the calculated discrete spectrum, the three evaluation indicators MSE, ARE, and RE are used for evaluation. To measure the stability and accuracy of the spectral reconstruction of the fitting method more accurately, it is necessary to perform 500 cycles of calculations to determine the corresponding MSE value and further analyze the influence of the fitting method on the reconstruction accuracy. The results show that different fitting methods should be adopted for different ground targets under the error conditions. The three indicators, MSE, ARE, and RE, have reached high accuracy and strong stability. The effect of high-precision reconstruction of the target spectrum is achieved. This article provides new ideas for related scholars engaged in hyperspectral reconstruction work and promotes the development of hyperspectral technology.

**Keywords:** spectral reconstruction; convex optimization; spectral fitting; sparse optimization

## 1. Introduction

In recent years, hyperspectral technology has been widely used in agriculture [1–4], resource exploration [5–8], oceanic studies [9], and environmental research [10]. Spectrometers are gradually becoming intelligent, miniaturized, and lightweight. At present, space-borne and airborne spectrometers must use miniaturized and high-precision spectrometers due to their volume limitations. At present, the main difficulties faced by miniature spectrometers are concentrated on two points. One is the influence of the number of filters in front of the detector on the accuracy of spectral information. The higher the number of filters, the higher the accuracy of the solution [11]. The second is to fit the

discrete spectral information. At present, few scholars have studied the spectral fitting method based on the filter. Thus, the focus of this article is to analyze the influence of the fitting method on spectral reconstruction.

The number of filters used in previous related research projects is about 200, and the spectral range is 400–900 nm. Therefore, the fitting method has little influence on the effect of spectral reconstruction. However, when the number of filters is small, the amount of data is sparse, and the fitting method has a greater impact on spectral reconstruction. Therefore, the fitting method is essential for a small spectrometer to achieve high-precision measurement. High-precision spectral deconstruction algorithms are the basis of high-precision spectral reconstruction. Many algorithms for spectral decomposition have recently been proposed.

Chang [12] analyzed the working process of the spectrometer and proposed a mathematical model for spectral reconstruction. Based on 200 filters, regularization and generalized cross-validation (GCV) were used to achieve high-precision spectral reconstruction. Finally, combined with non-uniform correction, the spectrum reconstruction accuracy is further improved, and the minimum value of the spectrum accuracy evaluation index ARE reaches 0.0248. In 2015, Bao [13] made innovations on the filter material, using quantum dot materials as 50 filters for spectral reconstruction, and achieved a better reconstruction effect.

In 2018, Zhang [14] proposed a reconstruction algorithm based on sparse optimization and dictionary learning based on 192 filters. The results show that the relative reconstruction error (RE) reaches 5.92%, achieving a good spectral reconstruction effect. In 2021, Zhao [15] used compressed sensing in the spectral reconstruction algorithm to reconstruct spectral reflectance. The experimental results prove that compressed sensing uses low-sampling data to achieve the effect of full sampling, which improves the accuracy of spectral reflectance reconstruction. The previous related research was based on hundreds of filters to reconstruct the target spectrum. Although a high reconstruction accuracy was achieved, hundreds of filters could not reconstruct the target spectrum due to the satellite spectrometer detector's volume limitation. This article uses 50 filters for analysis, which effectively reduces the amount of data and should prove vital for developing space-borne spectrometers.

## 2. Spectral Reconstruction Process

The reconstruction process is shown in Figure 1. The target spectrum first passes through the filter at the front of the detector. The energy on each filter can be obtained after the detector is modulated. The whole process can be considered the integration of the target spectrum, the transmittance of the filter, and the quantum efficiency of the detector within the wavelength range.

The integral expression is shown in Formula (1) [16,17]. $T_i(\lambda)$ is the transmittance function, $f(\lambda)$ is the detector quantum efficiency, $X(\lambda)$ is the target spectral function, $\lambda_1$ and $\lambda_2$ are the integral wavelength ranges, and $n$ is the number of filters.

$$\int_{\lambda_1}^{\lambda_2} T_i(\lambda)f(\lambda)X(\lambda)d\lambda, i = 1, 2, \cdots\cdots, n \tag{1}$$

The transmittance curve of the filter with detector characteristics in Figure 2 is the curve obtained by integrating the transmittance curve of the filter and the quantum efficiency curve of the detector in the range of $\lambda_1$ and $\lambda_2$, which is $A(\lambda) = T_i(\lambda)f(\lambda)$. In this paper, $\lambda_1$ and $\lambda_2$ are, respectively, 400 nm and 900 nm. By dividing the wavelength range of 400–900 nm equally, finding the area of each band, and then doing the same division on the target spectrum curve, it yields the average value of energy in each band. After discretization, the area of each band in each filter is multiplied by the mean value in the corresponding band of the target spectrum to obtain the energy $B$ of the target on each filter. In order to ensure the uniqueness of the solution of the discrete equation, matrix A is a square matrix of $n \times n$ (that is, the number of filters is equal to the dimension of matrix $A$).

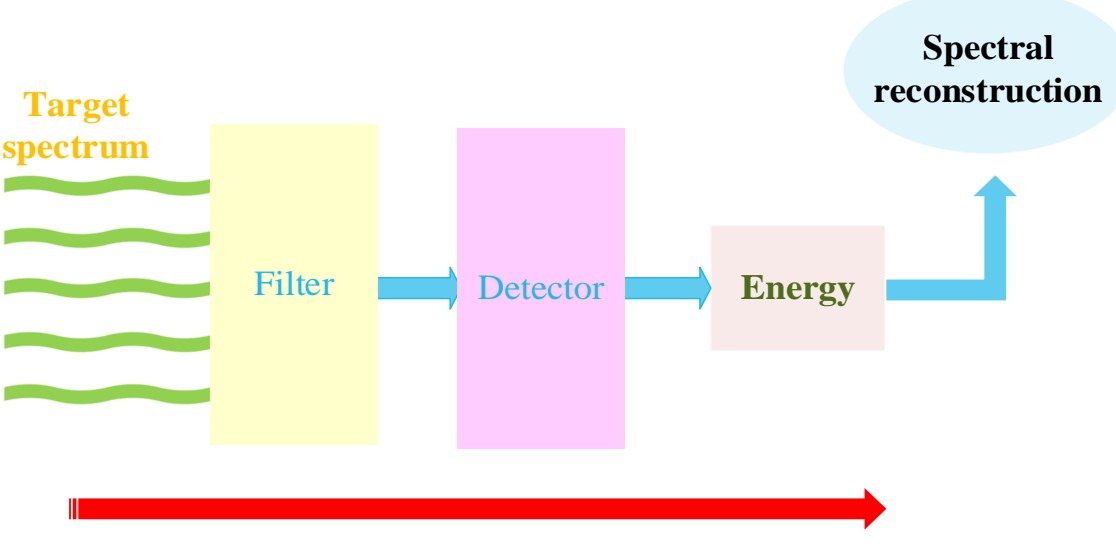

**Figure 1.** Working principle diagram of a spectrometer.

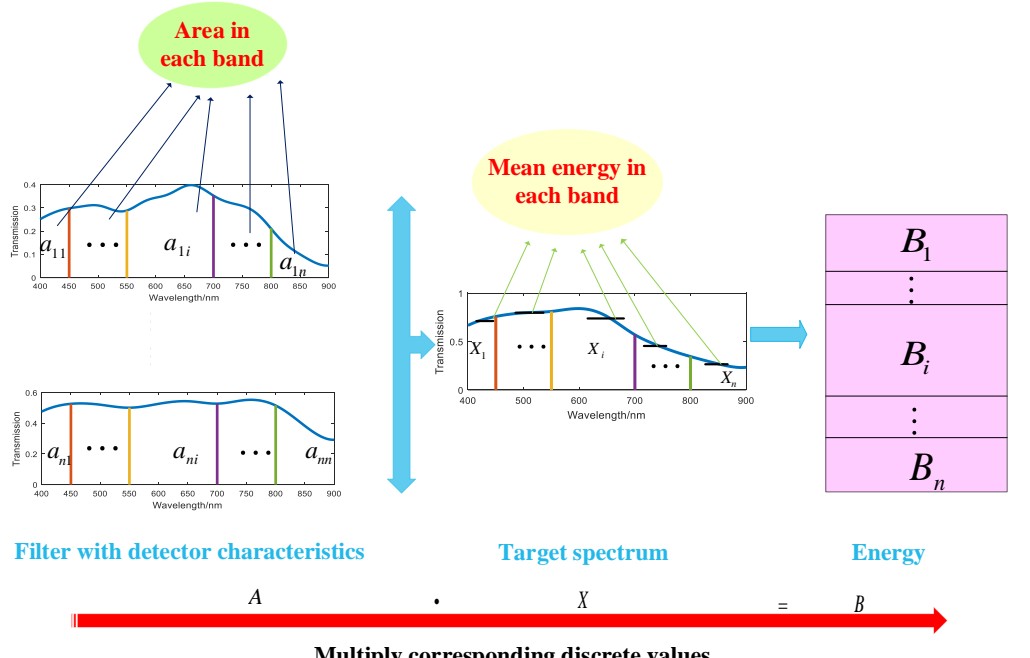

**Figure 2.** Discretization mathematical model of the spectral imaging system.

The discretized integral Equation (1) becomes Equation (2) [18].

$$A(\lambda) \cdot X(\lambda) = B \tag{2}$$

Expanding Equation (2) yields Equation (3).

$$\begin{bmatrix} a_{11} & a_{12} & \cdots & a_{1n} \\ a_{21} & a_{22} & \cdots & a_{2n} \\ \vdots & \vdots & \ddots & \vdots \\ a_{n1} & a_{n2} & \cdots & a_{nn} \end{bmatrix} \cdot \begin{bmatrix} X_1 \\ X_2 \\ \vdots \\ X_n \end{bmatrix} = \begin{bmatrix} B_1 \\ B_2 \\ \vdots \\ B_n \end{bmatrix} \tag{3}$$

However, the spectrometer's accuracy will be affected by a variety of error sources. The main error sources are the stray light error [19–21] and the detector non-uniformity error [22–24]. These errors make the equation *AX=B* unsuitable and add difficulty obtaining a solution for the equation.

In an ideal situation, the number of discrete wavelengths of the target spectrum should be equal to the number of filters. The greater the number of discrete wavelengths, the higher the spectral resolution.

To calculate the spectral information more accurately, we first divide the spectral curve of the filter into 25 evenly equal regions and calculate the average light intensity in the range of every 20 nm $Y_j(j = 1, 2, \cdots\cdots, 25)$. The spectral curve of the filter is divided into 50 equal parts (one part for every 10 nm), and the average light intensity $X_i(i = 1, 2, \cdots\cdots, 50)$ is calculated in the range of every 10 nm.

The theoretical expressions of $Y_j(j = 1, 2, \cdots\cdots, 25)$ and $X_i(i = 1, 2, \cdots\cdots, 50)$ are as shown in Equation (4). The calculation of $Y_j$ and $Y_j$ is solved by a convex optimization algorithm, and the specific expressions are shown in Equations (5) and (6):

$$(X_i + X_{i+1}) \approx 2Y_j, i = 1, 3, 5, \cdots\cdots, 49, j = 1, 2, \cdots\cdots, 25 \tag{4}$$

$$\begin{cases} \min||A_j \cdot Y - B_j||_2 \\ s.t. |A_j \cdot Y - B_j| \leq K' \\ |\frac{(Y_1 + Y_2 +, \cdots\cdots, + Y_{25})}{500} \times 20 - mean| < T' \end{cases} \quad j = 1, 2, \cdots\cdots, 25 \tag{5}$$

$$\begin{cases} \min||A_i \cdot X - B_i||_2 \\ s.t. |A_i \cdot X - B_i| \leq K \\ |\frac{(X_1 + X_2 +, \cdots\cdots, + X_{50})}{500} \times 10 - mean| < T \\ |\frac{X_i + X_{i+1}}{2} - Y_j| < Q \end{cases} \quad i = 1, 2, \cdots\cdots, 49, j = 1, 2, \cdots\cdots, 25 \tag{6}$$

In Equations (5) and (6), $X_i$ ($i = 1, 3, 5, \cdots\cdots, 50$) and $Y_j(j = 1, 2, \cdots\cdots, 25)$ are energy constraints. $Y_j$ is the estimated value calculated by Equation (5)

When Equation (1) is not discretized, calculate the average energy *mean* of the target spectrum in the range of 400–900 nm. $T'$ and $T$ are used as constraints to solve the overall mean value of the discrete spectrum. The calculated $Y_j$ ($j = 1, 2, \cdots\cdots, 25$) value is the average value of energy in every 20 nm wavelength range, and the calculated $X_i$ ($i = 1, 2, \cdots\cdots, 50$) value is the average value of energy in every 10 nm wavelength range.

The $Q$ in Equation (6) is the constraint between $X$ and $Y$. After obtaining the discrete values, they are considered the energy value of the center wavelength in the corresponding band. The data are then fitted to obtain the target reconstructed spectrum curve.

The choice of the filter is critical as it directly affects the accuracy of Equation (3). According to the matrix analysis, the condition number $cond(A)$ of the matrix $A$ composed of the filter is as small as possible to ensure the matrix equation is more robust [25]. In short, when designing the transmittance curve of the filter, the shape of the curve should have a low similarity. Based on this principle, the filter is designed and simulated.

## 3. Fitting the Original Spectra of Ground Objects

Since spacecraft functions are different, the observation targets are also different, but the spectral curve shapes of the same types of ground targets are similar. Therefore, the monochromatic light source and five typical objects are selected for spectral reconstruction. First, the target spectrum is normalized, and then the commonly used Gaussian, polynomial, and Fourier methods are selected for error-free fitting. The fitting results are shown in Figures 3–8.

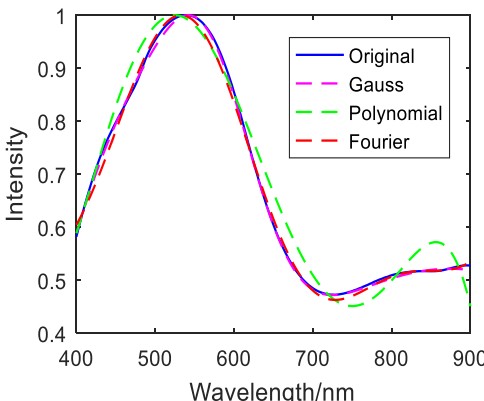

**Figure 3.** Fitted image of copper metal.

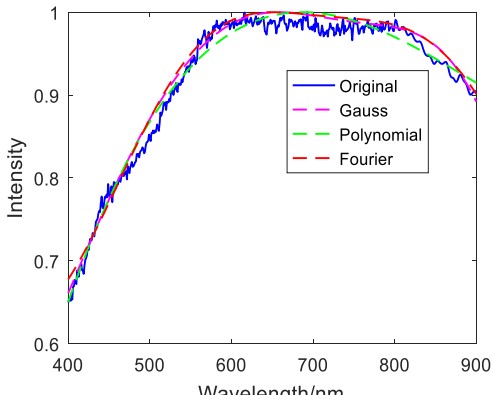

**Figure 4.** Fitted image of mica schist.

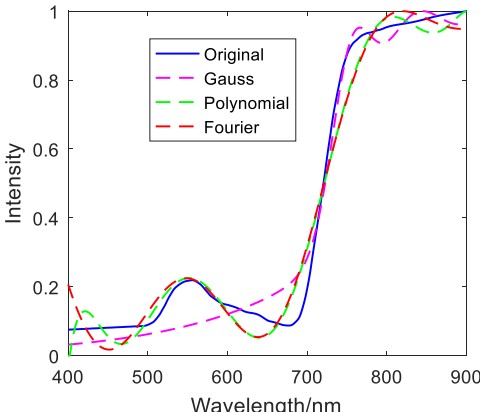

**Figure 5.** Fitted image of green plants (grass).

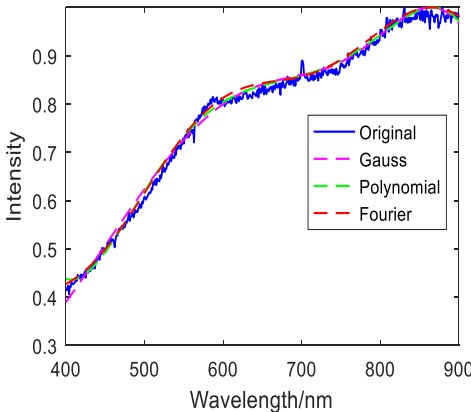

**Figure 6.** Jasper Ridge gravel fitting image.

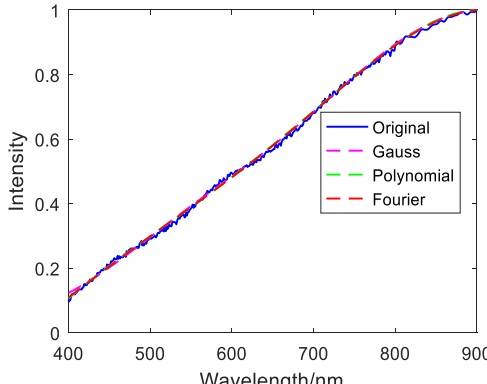

**Figure 7.** Fitting image of loam.

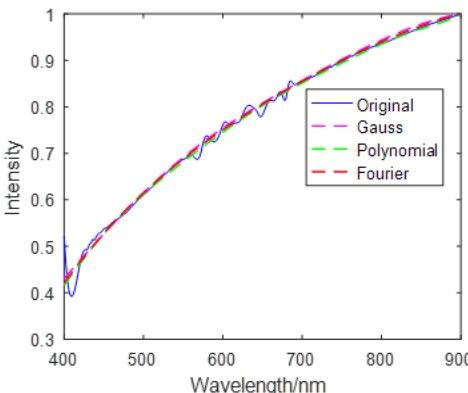

**Figure 8.** Asphalt fitting image.

There are different indicators for evaluating the effect of spectral reconstruction. Most previous studies have adopted MSE, ARE, and RE. The expressions of MSE, ARE, and RE are shown in (6)–(8). Equations (6) and (7) have similar meanings, but for better comparison with previous research, this paper uses the three indicators of MSE, RE, and ARE to evaluate the reconstructed spectrum. The smaller the three indicators are, the better the spectral reconstruction effect will be.

$$ARE = \frac{||y_i - \hat{y}||_2^2}{||y_i||_2^2} \tag{7}$$

$$RE = \frac{||y_i - \hat{y}||_2}{||y_i||_2} \tag{8}$$

$$MSE = \frac{1}{n}\sum_{k=1}^{n}(y_i - \overset{\wedge}{y})^2 \tag{9}$$

In the above equations, $y_i$ is the original target spectrum, $\overset{\wedge}{y}$ is the target spectrum after fitting, and $\overline{y}$ is the average value of the original spectrum. The specific evaluation indicators are shown in Tables 1–3.

**Table 1.** Gaussian fitting results of the original spectrum.

|  | MSE | ARE | RE |
|---|---|---|---|
| Copper metal | $2.4880 \times 10^{-5}$ | $5.0388 \times 10^{-5}$ | 0.0071 |
| Mica schist | $1.4671 \times 10^{-4}$ | $1.7104 \times 10^{-4}$ | 0.0131 |
| Grass | 0.0035 | 0.0107 | 0.1034 |
| Loam | $7.9708 \times 10^{-5}$ | $1.9231 \times 10^{-4}$ | 0.0139 |
| Jasper Ridge gravel | $1.9713 \times 10^{-4}$ | $3.0797 \times 10^{-4}$ | 0.0175 |
| Asphalt | $1.3364 \times 10^{-4}$ | $2.2575 \times 10^{-4}$ | 0.0150 |

**Table 2.** Polynomial fitting results of the original spectra.

|  | MSE | ARE | RE |
|---|---|---|---|
| Copper metal | 0.0010 | 0.0021 | 0.0462 |
| Mica schist | $1.5045 \times 10^{-4}$ | $1.7555 \times 10^{-4}$ | 0.0132 |
| Grass | 0.0025 | 0.0077 | 0.0879 |
| Loam | $7.4197 \times 10^{-5}$ | $1.7902 \times 10^{-4}$ | 0.0245 |
| Jasper Ridge gravel | $1.6368 \times 10^{-4}$ | $2.5553 \times 10^{-4}$ | 0.0160 |
| Asphalt | $1.0546 \times 10^{-4}$ | $1.7816 \times 10^{-4}$ | 0.0133 |

**Table 3.** Fourier fitting results of the original spectrum.

|  | MSE | ARE | RE |
|---|---|---|---|
| Copper metal | $1.2568 \times 10^{-4}$ | $2.5342 \times 10^{-4}$ | 0.0159 |
| Mica schist | $1.8001 \times 10^{-4}$ | $2.0959 \times 10^{-4}$ | 0.0145 |
| Grass | 0.0031 | 0.0097 | 0.0984 |
| Loam | $6.4220 \times 10^{-5}$ | $1.5494 \times 10^{-4}$ | 0.0126 |
| Jasper Ridge gravel | $1.8638 \times 10^{-4}$ | $2.9076 \times 10^{-4}$ | 0.0196 |
| Asphalt | $9.7617 \times 10^{-4}$ | $1.6491 \times 10^{-4}$ | 0.0128 |

Tables 1–3 show that copper metal and mica schist use Gaussian fitting to achieve the best results. Grass green plants, Jasper Ridge gravel, and asphalt have the best polynomial fitting effect. Loam uses Fourier fitting the best. The evaluation indicators when the best fit method is adopted for the six targets are shown in Table 4.

**Table 4.** Evaluation index of optimal fitting of target spectrum.

|  | Fitting Method | MSE | ARE | RE |
|---|---|---|---|---|
| Copper metal | Gaussian fitting | $2.4880 \times 10^{-5}$ | $5.0388 \times 10^{-5}$ | 0.0071 |
| Mica schist | Gaussian fitting | $1.4671 \times 10^{-5}$ | $1.7104 \times 10^{-4}$ | 0.0131 |
| Grass | Polynomial fitting | 0.0025 | 0.0077 | 0.0879 |
| Loam | Fourier fitting | $6.4220 \times 10^{-5}$ | $1.5494 \times 10^{-4}$ | 0.0126 |
| Jasper Ridge gravel | Polynomial fitting | $1.6368 \times 10^{-4}$ | $2.5553 \times 10^{-4}$ | 0.0160 |
| Asphalt | Polynomial fitting | $1.0546 \times 10^{-4}$ | $1.7816 \times 10^{-4}$ | 0.0133 |

## 4. Spectral Fitting of Discrete Objects with Errors

After determining the best fitting method of the original target spectrum, 50 filters are used to establish the ***AX=B*** equation set, and a 1% random error is applied. The convex optimization algorithm is used to calculate 50 discrete spectra, and fitting is performed. For the case of applied error, the best fitting method of each discrete target spectrum was determined. The error between the MSE value of each 10 nm wavelength and the standard value was calculated by comparing the changes of the fitting methods under the two conditions. The results are shown in Figures 9–20.

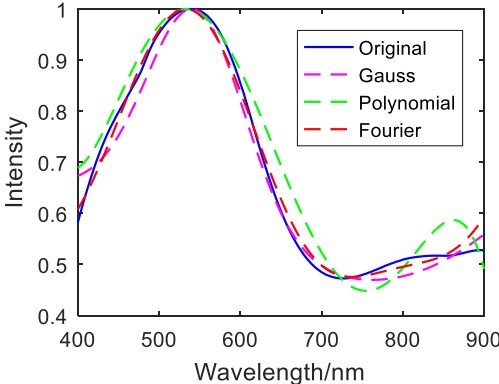

**Figure 9.** Fifty discrete value fitting images of copper metal.

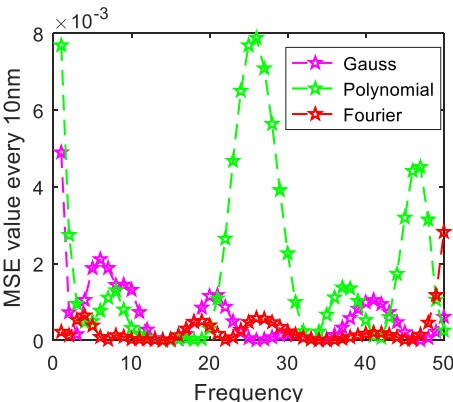

**Figure 10.** The distribution of MSE value error of every 10 nm of copper metal.

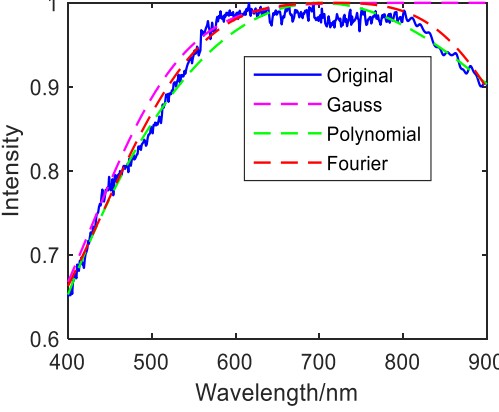

**Figure 11.** Fifty discrete value fitting images of mica schist.

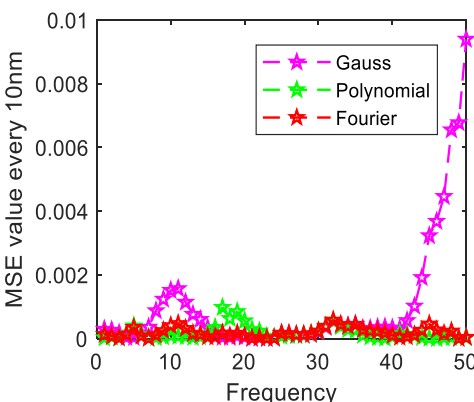

**Figure 12.** The distribution of MSE value error every 10 nm of mica schist.

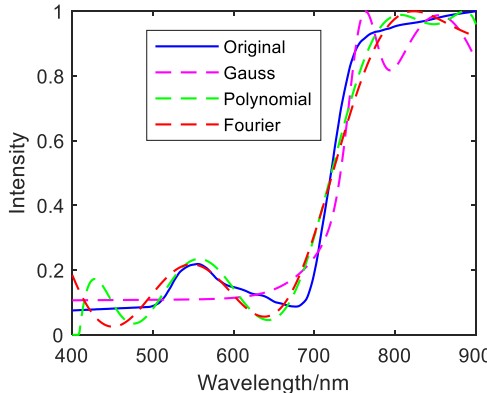

**Figure 13.** Fifty discrete value fitting images of green plants (grass).

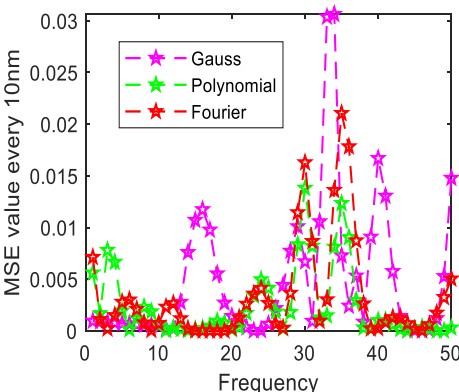

**Figure 14.** The distribution of MSE value error every 10 nm of green plants (grass).

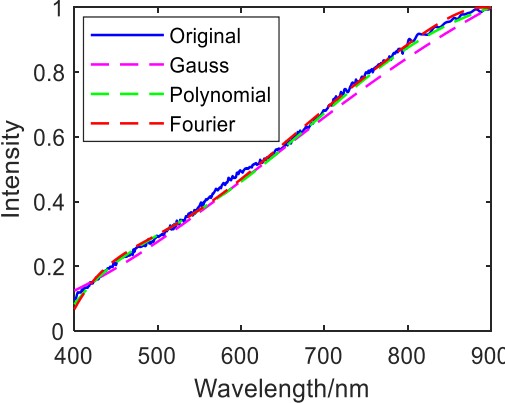

**Figure 15.** Fifty discrete value fitting images of loam.

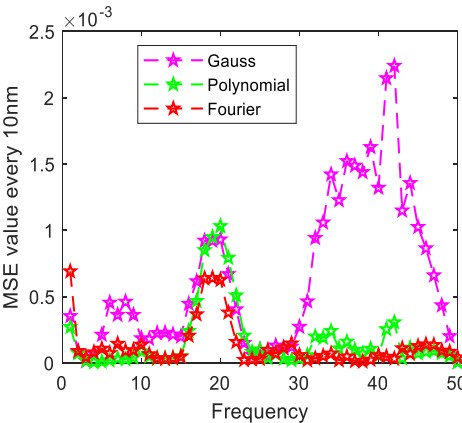

**Figure 16.** The distribution of MSE value error every 10 nm of loam.

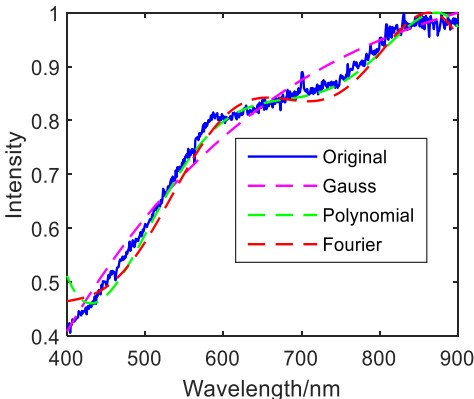

**Figure 17.** Fifty discrete value fitting images of Jasper Ridge gravel.

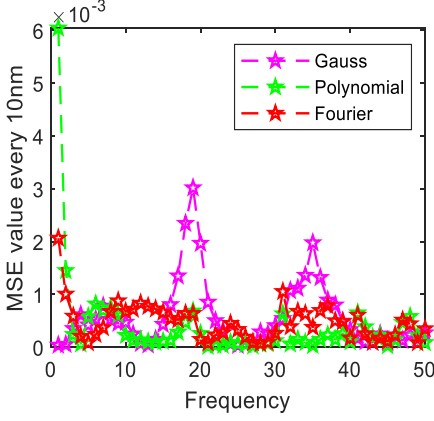

**Figure 18.** The distribution of MSE value error every 10 nm of Jasper Ridge gravel.

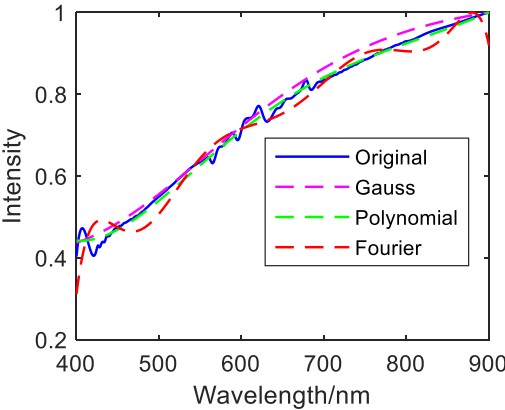

**Figure 19.** Fifty discrete value fitting images of asphalt.

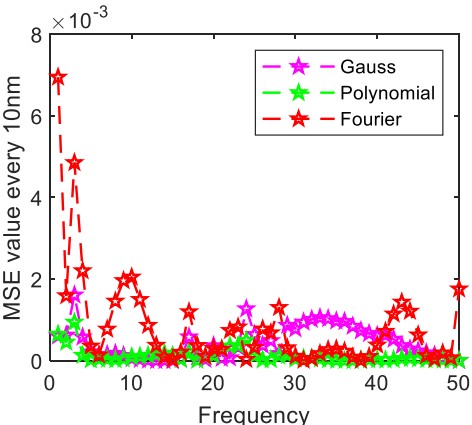

**Figure 20.** The distribution of MSE value error every 10 nm of asphalt.

The target spectrum curve fitting evaluation index is shown in Tables 5–7.

**Table 5.** Gaussian fitting evaluation index.

|  | MSE | ARE | RE |
|---|---|---|---|
| Copper metal | $6.2831 \times 10^{-4}$ | 0.0013 | 0.0356 |
| Mica schist | 0.0010 | 0.0012 | 0.0352 |
| Grass | 0.0050 | 0.0154 | 0.1243 |
| Loam | $6.4275 \times 10^{-4}$ | 0.0016 | 0.0394 |
| Jasper Ridge gravel | $5.8079 \times 10^{-4}$ | $9.0730 \times 10^{-4}$ | 0.0301 |
| Asphalt | $4.3241 \times 10^{-4}$ | $7.3048 \times 10^{-4}$ | 0.0270 |

**Table 6.** Evaluation index of polynomial fitting.

|  | MSE | ARE | RE |
|---|---|---|---|
| Copper metal | 0.0019 | 0.0038 | 0.0614 |
| Mica schist | $1.7523 \times 10^{-4}$ | $2.0402 \times 10^{-4}$ | 0.0143 |
| Grass | 0.0024 | 0.0075 | 0.0865 |
| Loam | $1.7204 \times 10^{-4}$ | $4.1509 \times 10^{-4}$ | 0.0204 |
| Jasper Ridge gravel | $3.8443 \times 10^{-4}$ | $5.9980 \times 10^{-4}$ | 0.0245 |
| Asphalt | $1.1884 \times 10^{-4}$ | $2.0076 \times 10^{-4}$ | 0.0142 |

**Table 7.** Fourier fitting evaluation index.

|  | MSE | ARE | RE |
|---|---|---|---|
| Copper metal | $2.6600 \times 10^{-4}$ | $5.5337 \times 10^{-4}$ | 0.0235 |
| Mica schist | $1.8525 \times 10^{-4}$ | $2.1573 \times 10^{-4}$ | 0.0147 |
| Grass | 0.0033 | 0.0102 | 0.1008 |
| Loam | $1.3221 \times 10^{-4}$ | $3.1900 \times 10^{-4}$ | 0.0179 |
| Jasper Ridge gravel | $4.7297 \times 10^{-4}$ | $7.3980 \times 10^{-4}$ | 0.0272 |
| Asphalt | $8.3006 \times 10^{-4}$ | 0.0014 | 0.0377 |

Tables 5–7 show that when there is an error, the discrete value calculated by *AX=B* is fitted. Mica schist, grass, Jasper Ridge gravel, and asphalt have the best fitting results by polynomial fitting, while copper metal and loam have the best fitting precision by Fourier fitting. The evaluation indicators are shown in Table 8.

**Table 8.** Evaluation index of optimal fitting of target spectrum with error.

|  | Fitting Method | MSE | ARE | RE |
|---|---|---|---|---|
| Copper metal | Fourier fitting | $2.6600 \times 10^{-4}$ | $5.5337 \times 10^{-4}$ | 0.0235 |
| Mica schist | Polynomial fitting | $1.7523 \times 10^{-4}$ | $2.0402 \times 10^{-4}$ | 0.0143 |
| Grass | Polynomial fitting | 0.0024 | 0.0075 | 0.0865 |
| Loam | Fourier fitting | $1.3221 \times 10^{-4}$ | $3.1900 \times 10^{-4}$ | 0.0179 |
| Jasper Ridge gravel | Polynomial fitting | $3.8443 \times 10^{-4}$ | $5.9980 \times 10^{-4}$ | 0.0245 |
| Asphalt | Polynomial fitting | $1.1884 \times 10^{-4}$ | $2.0076 \times 10^{-4}$ | 0.0142 |

Tables 5–7 show that the target polynomial fitting of mica schist, grass, loam, asphalt, and Jasper Ridge gravel has little difference with Fourier fitting, and the indicators are relatively close.

To further verify the correctness and stability of the above-mentioned feature target evidence fitting method, 500 calculations and fittings are performed, and the MSE value of each target under different fitting methods is calculated. The MSE distribution is shown in Figures 21–38.

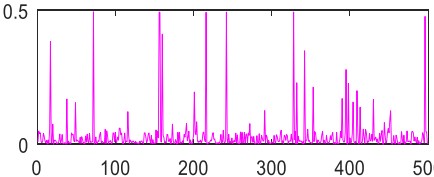

**Figure 21.** Gaussian fitting of copper metal MSE value distribution.

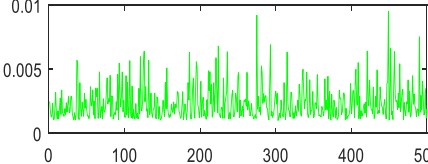

**Figure 22.** Polynomial fitting of copper metal MSE value distribution.

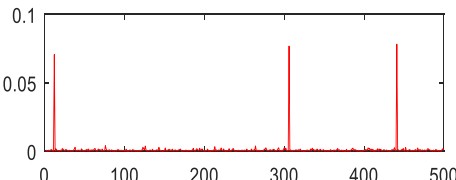

**Figure 23.** Fourier fitting of copper metal MSE value distribution.



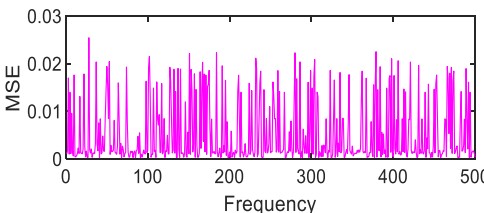

**Figure 24.** Gaussian fitting of mica schist MSE value distribution.

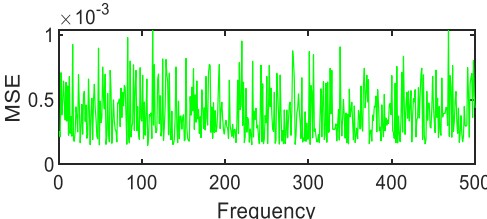

**Figure 25.** Polynomial fitting of mica schist MSE value distribution.

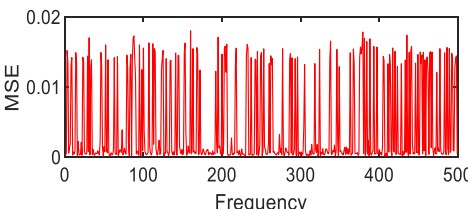

**Figure 26.** Fourier fitting of mica schist MSE value distribution.

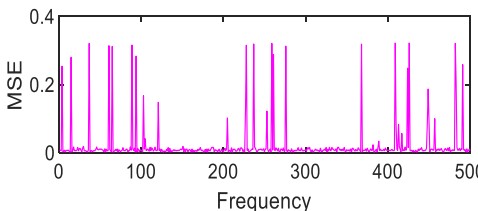

**Figure 27.** Gaussian fitting of grass MSE value distribution.

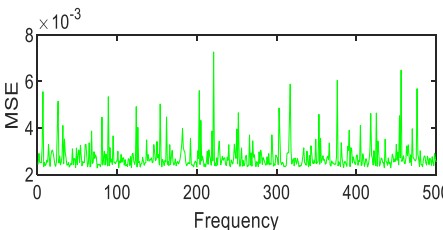

**Figure 28.** Polynomial fitting of grass MSE value distribution.

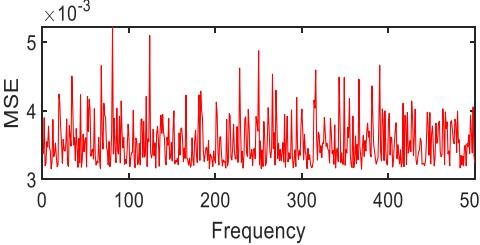

**Figure 29.** Fourier fitting of grass MSE value distribution.

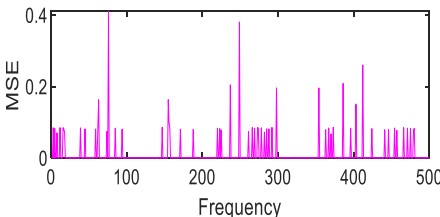

**Figure 30.** Gaussian fitting of loam MSE value distribution.

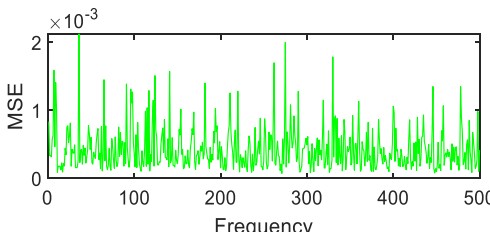

**Figure 31.** Polynomial fitting of loam MSE value distribution.

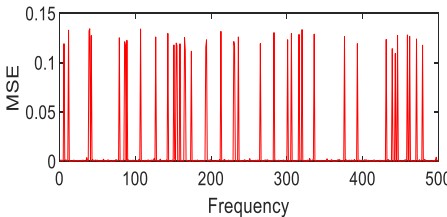

**Figure 32.** Fourier fitting of loam MSE value distribution.

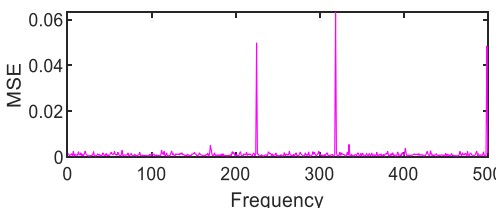

**Figure 33.** Gaussian fitting of Jasper Ridge gravel MSE value distribution.

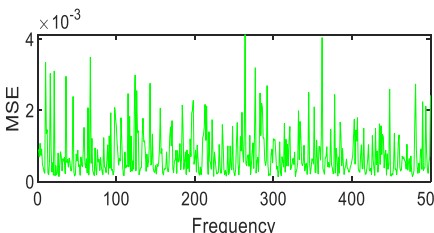

**Figure 34.** Polynomial fitting of Jasper Ridge gravel MSE value distribution.

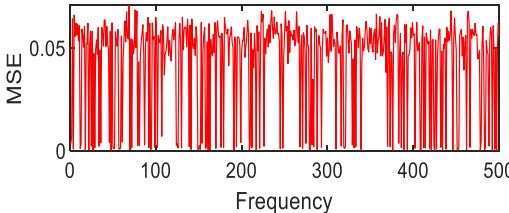

**Figure 35.** Fourier fitting of Jasper Ridge gravel MSE value distribution.

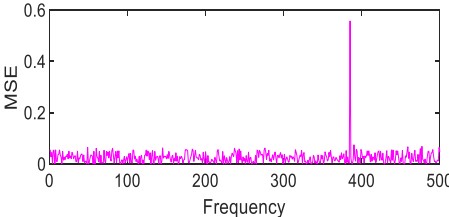

**Figure 36.** Gaussian fitting of asphalt MSE value distribution.

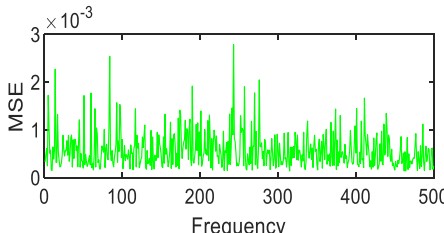

**Figure 37.** Gaussian fitting of asphalt MSE value distribution.

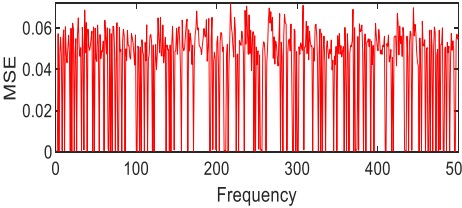

**Figure 38.** Fourier fitting of asphalt MSE value distribution.

After 500 calculations under the condition of 1% random error, the following results can be obtained. Copper metal, mica schist, loam, and Jasper Ridge gravel achieve polynomial fitting accuracy with the highest MSE average values of 0.01, $1 \times 10^{-3}$, $2 \times 10^{-3}$, $4 \times 10^{-3}$, respectively. The fitting accuracy of Fourier and polynomials of grasses and other green plants is about 10e-3, and the effect of Fourier fitting is better. In particular, it is pointed out that although Gaussian fitting has higher accuracy, the MSE value is prone to jump, so Gaussian fitting is not suitable for the above-mentioned object targets.

## 5. Conclusions

In this study, the working principle and mathematical model of the spectrometer are elaborated. Six typical targets are selected based on fifty filters. The convex optimization algorithm is used to solve the Equations (2). Unlike the previous research, the number of filters used in this paper is fewer, and the spectral data are sparser, so the fitting method is critical. Based on the 50 transmittance curves given in this paper, the best fitting method of the original spectrum and the 50 discrete spectrums obtained by solving Equation (2) re analyzed. In terms of spectral reconstruction evaluation, the three indicators ARE, RE, and MSE are used for evaluation, further improving the accuracy of the spectral reconstruction evaluation. The convex optimization solution method, fitting method, and spectrum evaluation index proposed in this study are unique and promising to advance the field of spectrum reconstruction and are also conducive to the miniaturization of the spectrometer.

The transmittance curve of 50 filters in the wavelength range of 400–900 nm is shown in Figure 39 below.

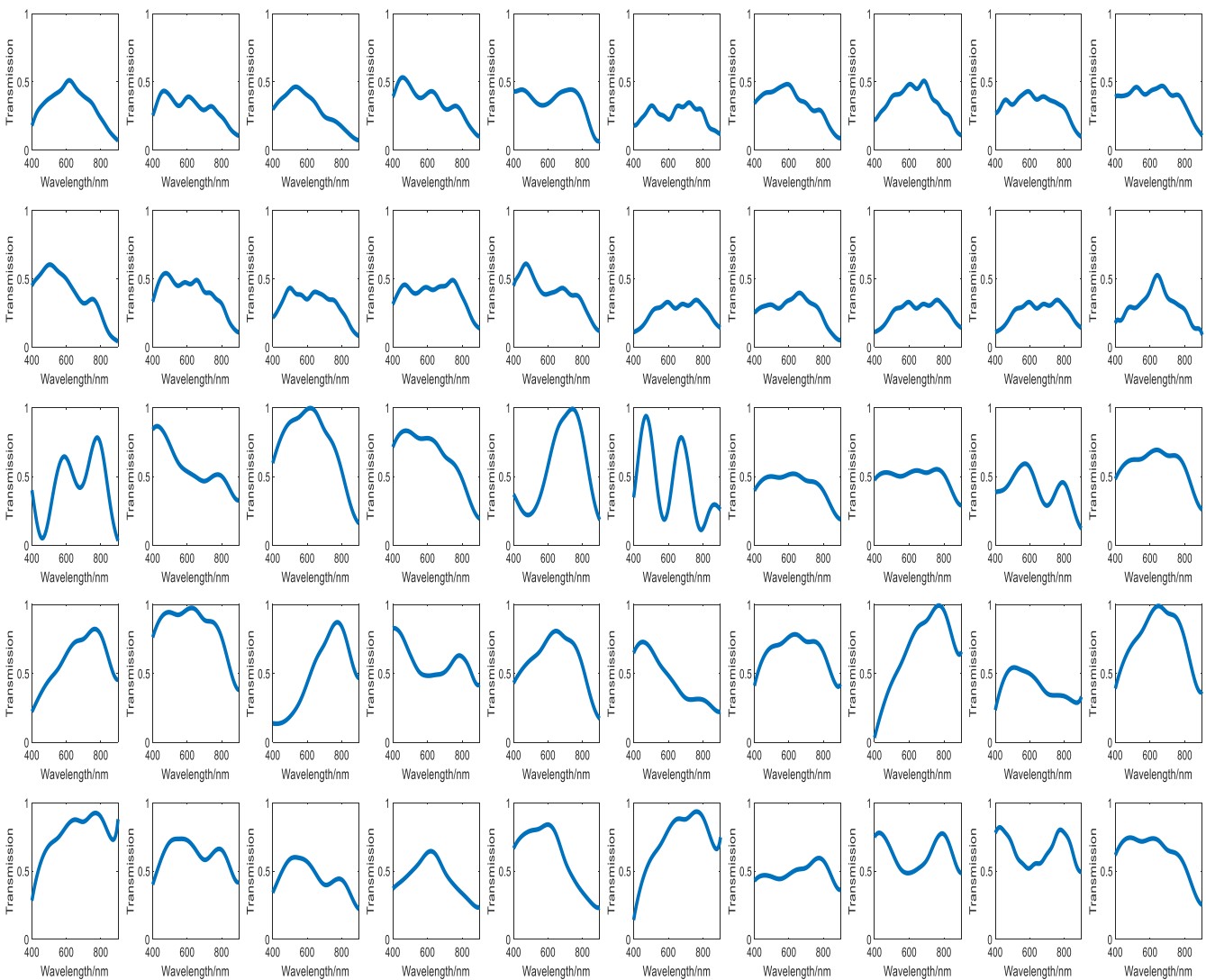

**Figure 39.** Transmittance curves of 50 filters.

**Author Contributions:** L.Z. and J.Z. conceived the prototype and designed the experiments; H.S. performed the experiments and analyzed the data; J.Z., H.S., W.Z. and W.W. guided the improvement of the algorithm and the analyses of the results. L.Z. and W.W. revised the manuscript. All authors have read and agreed to the published version of the manuscript.

**Funding:** This research received no external funding.

**Institutional Review Board Statement:** Not applicable.

**Informed Consent Statement:** Not applicable.

**Data Availability Statement:** Not applicable.

**Conflicts of Interest:** The authors declare no conflict of interest.

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
