# Peer review of "Case Study on the Fitting Method of Typical Objects"

_photonics, doi:10.3390/photonics8100432_

Round 1

Reviewer 1 Report

The present ms investigates spectral reconstruction methods for different targets. I am not familiar with the substantive aspect of the study. Hence, I can only evaluate the precision and clarity of the description of the mathematical methods used in the ms and the presentation style.

My comments are as follows:
1.    The title of the paper sounds a bit peculiar. Maybe substitute ”Research on …“ with ”An investigation on …“ or ”A case study on …“.
2.    Abstract, line 10: What is an ”equation group“?
3.    Abstract and ms: Why should there be an error of 1% being added? Authors should justify this. Why 1% and not any other percentage value?
4.    Sect. 2: It should be better explained which data has been observed and what should be mentioned. Please mention this clearly and refer to mathematical symbols.
5.    P. 3: Equation (1) is not an integral equation; it is just an integral. So, I cannot figure out which integral equation is approximated by Equation (2).
6.    Also, define A(\lambda) based on the introduced notation for Equation (1).
7.    l. 107 and other lines: write ”j“ in italic. 
8.    In addition, mathematical notation on l. 113 in a larger font.
9.    The style of Equations (5) and (6) needs improvement. First, write ”minimize“ without space and in non-italic. Also, mention which of the variables is estimated, I suppose it is Y, but this should be clearly stated. Overall, I think that the estimation problem discussed in Sect. 2 is relatively simple, but the authors described it in a way that, for an outsider of the research area it is nearly impossible to understand
10.    Section 3 and 4 contain a somewhat lackluster compilation of figures and tables. The authors could think about an improved and more condensed presentation style.
11.    Fit criteria Equations (7), (8), (9): The ARE is just the square of RE so that it can be removed from the tables. Moreover, consider reporting RMSE (square root of MSE) instead of MSE for reporting because it avoids printing too small numbers. Moreover, which additional information is provided by reporting MSE (or RMSE) in addition to RE?
12.    Equation (9): How are the weights \nu_i be defined?
13.    Would it be helpful to regularize the estimated functions (e.g., by including roughness penalties) obtained from the discretized integral equation? This could also further stabilize estimation. 

Author Response

Thank you very much for reviewing our article. We feel that your questions about the details of the article are very helpful to our article 

Question 3:Why should there be an error of 1% being added? Authors should justify this. Why 1% and not any other percentage value?

The 1% error is determined by the nature of the detector. We have done some experiments and found that the gray value of each band is within 1%.

Question 4:It should be better explained which data has been observed and what should be mentioned. Please mention this clearly and refer to mathematical symbols

Question 11: Fit criteria Equations (7), (8), (9): The ARE is just the square of RE so that it can be removed from the tables. Moreover, consider reporting RMSE (square root of MSE) instead of MSE for reporting because it avoids printing too small numbers. Moreover, which additional information is provided by reporting MSE (or RMSE) in addition to RE?

Reply to questions 4 and 11:Since the number of articles engaged in related research is very small, the number is less than 10, but these articles have three evaluation methods for reconstruction accuracy: MSE, ARE, and RE. When writing the article, we took into account that these three evaluation methods may be There is overlap, but in order to better evaluate the reconstruction effect, we hope to adopt all the previous evaluation standards, so that it can give the relevant staff a clearer expression.

Question 13:Would it be helpful to regularize the estimated functions (e.g., by including roughness penalties) obtained from the discretized integral equation? This could also further stabilize estimation.

The work on regularization has been proposed by related scholars before, but in fact the effect is not good. We have improved the method based on previous research. The application of regularization in spectral reconstruction can be found in Reference 12.

The remaining questions have been corrected and marked in the article, please read the revised version

Reviewer 2 Report

The paper describes and compares three different approaches (polynomial, Gaussian and Fourier fittings) for spectral reconstruction based on the data coming from 50 filters.

The major concerns regarding the paper is that none of the above mentioned algorithms can capture correctly the spectral shape.

Over the three presented algorithms, only the Fourier fitting has the capacity of presenting a sufficient "elasticity" to capture the curvatures of the spectra.

Therefore, I wonder why the authors did not try more relevant fitting algorithms such as the regression splines, the assymetric least squares or even the wavelets that, according to the litterature, would be more sounded for this application.

Another point relates to the presence of a weight parameter in the MSE formula. What is this weight and how is it determined? I never saw a weight factor in the MSE computation. Please explain.

Author Response

Thank you very much for reviewing our article. We think your review is very helpful to the improvement of our article quality .

The fitting effect of the spline regression method is very good in most cases. The MSE, ARE, and RE indicators all meet the requirements, but we found that the fitting accuracy may be poor when the error was added for 500 cycles. Before experimenting with a monochromatic light source and using many different fitting methods, we found that Gaussian, polynomial, and Fourier are the most stable .

In response to question 2, we have corrected it in the article.

Round 2

Reviewer 1 Report

The revised ms addressed most of my concerns. 
Minor points:
1.    Typo in Eq. (5) and (6): write ”minimize“ instead ”minmize“

Reviewer 2 Report

The authors have replied to all my concerns

This manuscript is a resubmission of an earlier submission. The following is a list of the peer review reports and author responses from that submission.